# Autologous Tooth Granulometry and Specific Surface Area with Three Grinding Methods: An In Vitro Study

**DOI:** 10.3390/ma17040773

**Published:** 2024-02-06

**Authors:** Berta Lavarias Ribes, Ignacio Fernández-Baca, Javier Gil Mur, Joaquin López-Malla Matute, Juan Manuel Aragoneses Lamas

**Affiliations:** 1Bioengineering Institute of Technology, Faculty of Medicine and Health Sciences, International University of Catalonia, Sant Cugat del Vallés, 08195 Barcelona, Spain; bertalavariasribes@gmail.com (B.L.R.); nfernandezbaca@gmail.com (I.F.-B.); xavier.gil@uic.es (J.G.M.); 2Faculty of Dentistry, Universidad Alfonso X El Sabio, 28037 Madrid, Spain; jmaragoneses@gmail.com; 3Department of Dental Research, Federico Henriquez y Carvajal University, Santo Domingo 11005, Dominican Republic

**Keywords:** particulate dentine, periodontal regeneration, autologous tooth, autograft, xenograft, alveolar preservation, dental extraction

## Abstract

A postextraction socket becomes a clinical challenge due to the fact that a series of changes associated with bone remodelling and resorption of the socket that occur after extraction, which limits the aesthetic and functional prognosis of implant-supported rehabilitations. It has been studied that the use of the autologous tooth-derived graft (ATDG) has regenerative properties and could therefore be useful for solving this type of problem. There is no consensus in the scientific literature on a standardized protocol for the use of the autologous tooth. Therefore, the aim of the present study was to evaluate the most relevant parameters to achieve the best properties of ground ATDG using three methods, namely Gouge forceps, electric grinder, and manual, that made up the study group (SG) and compared with the control group (CG) consisting of Bio-Oss^®^. The sample obtained by the electric grinder had the highest value of specific surface area (2.4025 ± 0.0218 m^2^/g), while the particle size as average diameter (751.9 µm) was the lowest and most homogeneous of the three groups. Therefore, the electric grinder allowed for obtaining ATDG with more regenerative properties due to its specific surface-area value and particle size in accordance with the xenograft with the greatest bibliographical support (Bio-Oss^®^). The higher specific surface increases the reaction with the physiological media, producing faster biological mechanisms.

## 1. Introduction

The alterations that occur at the hard-tissue level are more evident in the first two or three months postextraction, being more evident in the vestibular cortex and giving the appearance of a displaced alveolar ridge in the lingual or palatal direction [1,2]. The resorption process begins immediately after extraction, with a 50% loss in the width of the alveolar process within three months, which is usually greater than in height [3], the alveolar socket undergoes an average horizontal shrinkage of between 0.9 and 3.8 mm and an average vertical reduction of 1.24 mm within three to seven months after tooth extraction [4]. Bone loss is significant in the first six months postextraction but continues slowly throughout life, depending on the individual [5].

These changes, which were first described by Amler [6], have been demonstrated in several studies, such as Araujo [3] and Cardaropoli [7], who investigated the healing dynamics of alveoli after extraction in a preclinical model. The healing of a postextraction socket begins with the formation of a clot that transforms into a connective tissue matrix that will calcify to form trabecular bone and eventually form mature cortical or trabecular bone.

The placement of an implant requires an adequate quantity and quality of bone. Sometimes, due to the aforementioned resorption, among other factors, this bone volume is insufficient, and subsequent augmentation techniques are required. Alveolar preservation is any technique that seeks to minimise horizontal and vertical bone loss of the ridge after extraction. They are a set of procedures performed at the time of tooth extraction with the aim of decreasing bone resorption while preserving alveolar dimensions and contours for future implant or conventional prosthesis placement [8]. To preserve the soft tissue, some authors [9] take a digital measurement prior to tooth extraction to copy the emergence profile and, once the implant is placed, take a new measurement that allows a provisional to be screwed in that duplicates the emergence profile that the tooth had before being extracted. The core of the customized abutment is manufactured from polyether ether ketone. It was observed how weeks after placement of the provisional the emergency profile is maintained.

These procedures involve less bone loss than the natural healing process, making implant placement possible and avoiding the need for more invasive guided bone-regeneration procedures in most cases compared to the natural healing of the socket [10]. 

Autogenous bone graft is considered the gold-standard biomaterial [11,12] in bone regeneration procedures due to its osteogenic, osteoinductive, and osteoconductive capacities. Despite these beneficial properties, we can also find some limitations, such as its availability, the increased patient morbidity involved in obtaining it, and its high resorption rate [13,14]. To overcome these limitations, bone substitutes have been proposed that are encompassed in allografts, xenografts, and alloplastic grafts, which, despite being different in composition and shape, can give similar clinical results.

A biomaterial that is used as a bone substitute should possess qualitative criteria, namely biocompatibility, which represents the capability of providing osseointegration without causing inflammatory reactions [15,16]; osteoconductivity, the natural properties that allow cell activity, reproduction and amplification; and osteoinductive properties, or a capability of triggering the biochemical and modulating processes, so stem cells can differentiate into osteoblasts, osteoclasts, and osteocytes and induce osteogenesis, which is related to the formation of a new bone matrix [17,18].

The grain size and composition of the biomaterial are considered to directly influence the activity and speed of resorption [19]. The main disadvantage of these biomaterials lies in their economic cost and lack of osteogenesis, as well as their limited osteoinductivity [20,21].

In this context, the human tooth itself is proposed as an autologous biomaterial, which is obtained at the time of extraction and could have osteogenic and osteoinductive properties potentially superior to those of other substitutes [22,23,24]. The biological plausibility of these properties lies in the similar composition of tooth and bone [25,26], a common embryonic origin, bone differentiation capacity in vitro [27], osteoconduction and osteoinduction in vivo [28,29,30], and excellent biocompatibility [31]. The dental matrix has a porous microstructure that promotes cell adhesion, blood circulation, and slow resorption that guarantees correct osteoconduction, thus ensuring that the bone volume is preserved for longer [32]. Furthermore, it has given good results in maintaining the length and width of the alveolar ridge after extractions [33]. If the dental matrix is processed in the right way, it has osteoinductive properties determined by the morphogenetic proteins BMPs, which are contained in physiological quantities in the tooth. The crushed tooth seems to preserve these osteoinductive properties with a partial demineralization of the dental matrix without damaging its structure [34]. This is why the use of this biomaterial is presented as a promising option in regenerative procedures.

However, the existing literature is still scarce and, at some points, contradictory [35], which could be due to the lack of standardised protocols in the processing of the tooth, affecting the properties of the biomaterial [36].

The protocols described in the literature differ depending on several points, such as the type of device used for processing, the granulometry of the material, the processing of the material obtained (disinfection, demineralisation treatment, etc.), or the inclusion or not of the enamel [21,24,37,38,39,40,41,42,43,44,45].

Among the mills used for tooth granulation, a distinction is made between mortar and pestle mills and automatic mills. Once crushed, it is possible to subdivide those protocols that include the use of sieves from those that do not, in order to differentiate the particle size [41,42,43,44,45,46,47,48,49,50].

Regarding dentine processing, it is possible to observe in the literature the use of the graft in different forms, native without treatment, disinfected, sterilised, demineralised, freeze-dried, and deproteinised, each of them with different concentrations and times of a wide range of chemical agents [41,42,43,44,45,46,47,48,49,50].

In view of the wide divergence of protocols used in the processing of the tooth as a regenerative material, it is necessary to study the most relevant parameters to achieve the best properties of the material called autologous tooth-derived graft (ATDG). For this purpose, the three ATDG crushing methods found in the literature were used arbitrarily in the different studies, which were:Gouge forceps from Helmut Zepf^®^ (Seitingen-Oberflacht, Germany) as a manual processor. It is a cutting instrument with two articulated branches with concave active ends and pointed ends, between which the tooth is placed. After applying pressure, the piece is fractured into several pieces, with the possibility of repositioning the pieces of teeth between the valves to make them smaller.Automatic Smart Dentin Grinder Kometa-Bio from Bioner^®^ (Madrid, Spain) as a processor with an automatic grinding device. The grinding time was three cycles of twenty seconds each at 240 Volts (V), with a frequency of 50 Herz (Hz) at 16,000 revolutions per minute (rpm). After grinding, the particle obtained was sieved for thirty seconds at 6 V at 10,000 rpm in the removable compartment. Dentin particles smaller than 1200 µm fell into the upper drawer, whereas the lower sieve filtered those smaller than 200 µm, which fell into the lower drawer for subsequent extraction. Screened ATDG was obtained for analysis because it was the only grinder that had a screen in its design;Manual grinder from Master Surgical SL^®^ (Madrid, Spain). A manual grinder was used for processing with a manual grinding device. The tooth was crushed by manually rotating the grinder twenty times until the particle size was as homogeneous as possible with respect to each other. This device does not have the option to sift the crushed sample.

## 2. Materials and Methods

### 2.1. Study Design

The characteristics of a biomaterial should be constant and precise; the fact that this biomaterial can be processed by different methods leads us to believe that there may be some discrepancy in the final product depending on the device used. To confirm or disprove this hypothesis, an evaluation of the morphology, granulometry, and specific surface area of ATDG processed by different manual and automatic systems was proposed.

After carrying out a systematic review of the existing scientific literature [41,42,43,44,45,46,47,48,49,50], different protocols for processing the autologous tooth to obtain the biomaterial were found, which can be divided into three groups, namely manual processing, processing with a manual grinding device, and processing with an automatic grinding device.

We proposed to analyse the products obtained from these processes by analysing the samples obtained with a [41,42,43,44,45,46,47,48,49,50]. The study group (SG) consisted of the ATDG, and the Control Group (CG) was the most scientifically supported xenograft [51,52], Bio-Oss^®^ from Geistlich Pharma AG (Wolhusen, Switzerland).

SG and CG were subdivided into three groups, consisting of the three grinding methods explained above, namely Gouge forceps (Group A), an Automatic Smart Dentin Grinder from Bioner^®^ (Madrid, Spain) (Group B), and a manual grinder from Master Surgical SL^®^ (Madrid, Spain) (Group C).

### 2.2. Sample Processing

The study included patients over the age of eighteen who needed dental extraction treatment and were in good health (ASA-1 and ASA-2). Tooth extractions were required for trauma, caries, or periodontal disease. Prior to extraction, patients signed an informed consent form, and the Ethics Committee of Alfonso X El Sabio University (Madrid, Spain) approved the use of these teeth for research purposes. 

The calculation of teeth used was shown to be at least 120 teeth. We have used for this study 149 teeth that have been extracted from 125 patients (*n* = 125). The calculation was carried out according to international standards to obtain the minimum quantity (30 g for each analysis) to carry out the studies of granulometry and specific surface area. The large number of patients homogenizes the differences in the different qualities of the teeth used.

Pregnant subjects, patients with a history of allergies, if they were current heavy smokers (more than 10 cigarettes/day), diabetes, cancer, human immunodeficiency virus (HIV), bone or metabolic diseases, immunosuppressive agents, or use of systemic corticosteroids or intramuscular/intravenous bisphosphonates, and patients in radio or chemotherapy were excluded. No teeth with a root-canal treatment were selected. 

After human tooth extraction, soft tissue debris, restorations, caries, and foreign bodies were removed with the help of turbine diamond burs, contra-angle tungsten carbide burs, and the Intensiv^®^ Perio set diamond bur kit (Collina d’Oro, Switzerland). Subsequently, the tooth was disinfected by immersing it in a 70% ethanol solution until processing. 

After drying, the teeth were ground for each grinding system. The experiment was repeated five times for each system to determine the result. The number of particles required was determined by the instrumentation used, which was standardized according to ISO: ISO 9277:2022 [53]. Determination of the specific surface area of solids was by the gas-adsorption BET method. ISO 2010. p. 24 and ISO 13318-2:2007 for the determination of particle size distribution by centrifugal liquid sedimentation methods ISO 2021. p. 13 [54]. 

The operation of the machinery used to obtain the results allowed for a certain amount of particle size for its operation, which was justified by the ISO standard.

Once the three groups had been processed, the specific surface area was studied, which is defined as the ratio between the total surface area per unit of mass and its units in m^2^/g. Therefore, the greater the specific surface area, the greater the possibility of interaction with living tissues the biomaterial will have, or, in other words, the greater the particle’s reactivity with the medium [55].

The analysis of the specific surface areas of Groups B and C was carried out with the help of the Micromeritics^®^ ASAP 2020 (Figure 1) (Unterschleißheim, Germany). Nitrogen was used as an adsorbate to keep the biomaterial retained on the surface. The samples used were degassed at 100 °C under vacuum conditions (10 µHg), and the specific surface area was analysed by applying mathematical calculations described by the BET (Brunauer–Emmett–Teller) theory [56].

The particle size analysis of the samples from the crushed group with the manual and electric grinder was carried out using a Beckman Coulter^®^ (Brea, CA, USA) model LS13 320 laser particle size analyser using the “Tornado” configuration and dry analysis, with a particle diameter measurement range between 0.04–2000 µm. The Malvern Panalytical® Mastersizer 3000 (Almelo, The Netherlands) liquid analysis equipment was not used due to the impossibility of adjusting the obscuration of the liquid medium between 5% and 10% as recommended by the equipment manufacturer and due to the problems of obstruction of the equipment’s aspiration ducts, as it was not possible to measure diameters greater than 1 mm. For these reasons, the samples were analysed with the dry analysis equipment.

This was carried out using a laser-diffraction technique to measure particle size by measuring the intensity of scattered light as a laser beam passed through the particulate sample. This data was analysed to calculate the size of the samples according to the scattering pattern.

Since Group A presented values that the high-sensitivity particle size equipment did not admit because they were particles larger than 2000 μm, the sample had to be analyzed randomly with a JEOL JSM 5410^®^ (Jeol, Akishima, Japan) scanning electron microscope (SEM) (Musashino, Akishima, Tokyo) performing a total of three thousand measurements of eighty-one particles with the Image J image-analysis system version 2.1 (Java, Tokyo, Japan) integrated with the SEM, between which the sample was shaken and deposited back on the plate to perform a measurement of each particle, obtaining an average of all the measurements performed and thus a value for its particle size. In Figure 2 we observe three captures obtained with the SEM of each of the groups at a 35 magnification.

An area analysis covering a large amount of the surface of the samples was performed, as well as one or more spot analyses, in order to carry out a complete study of the surfaces. After sputtering with gold, the samples were analysed in the SEM using a voltage acceleration of 10 kV for imaging and 15 kV for EDS microanalysis. The SEM images were taken at ×50, ×1000, and ×3500 magnification.

Once the samples were obtained and categorised by granulometry, they were compared with each other and with the xenograft Bio-Oss^®^ from Geistlich Pharma AG (Wolhusen, Switzerland), a biomaterial of bovine origin derived from porous cancellous bone, where all the native organic material is eliminated by a chemical extraction process at low temperature (300 °C), maintaining the physical architecture intact. It consists of an interconnected porous system that favours the initial stability of the clot and subsequent growth of blood vessels within it, thus promoting the migration of osteoblasts and, thus, the formation of new bone [57,58].

According to international standards for particle size and specific surface-area analysis, at least 30 g should be obtained for each method studied. In the present study, 49 g were analyzed for the Gouge forceps, 47 g for the smart dentin grinder, and 51 g for the manual grinder. The particles corresponding to each method exceeded 6 million particles for each method. The particle size and specific surface area were analyzed by the equipment by obtaining Gaussian bells of the specific sizes and surfaces. For this reason, it was not possible to make a sample calculation or statistical studies since the standard indicates how to do the study, and its statistical value is the Gaussian curves, where the number of particles for each size interval and specific surface can be seen (ISO 2010. p. 24, ISO 13318-2:2007, ISO 2021. p. 13).

In this way, we checked whether the material obtained by the different processing systems was homogeneous or had discrepancies and, thus, compared the particle size with one of the most widely used and scientifically tested biomaterials.

## 3. Results

### 3.1. Specific Surface Area

#### 3.1.1. ATDG

The specific surface area of group A was 1.7451 ± 0.0183 m^2^/g; large particles were obtained due to the fact that the gouge is not able to grind uniformly and create particles of small size and homogeneous between them, as it only has two blades between which the sample is placed to exert operator-dependent force from the handle to grind it.

The specific surface area for Group B was 2.4025 ± 0.0218 m^2^/g, which was the highest value among the three grinding systems, possibly due to the protocol used to obtain it. The grinding time with the electric grinder was three cycles, after which the sample was sieved to filter the already-ground particles in the removable compartment. The dentine particles smaller than 1200 µm fell into the upper drawer where the lower sieve filtered the particles smaller than 200 µm, which fell into the lower drawer for further elimination, thus obtaining a sample of small size and with particles that were regular with each other.

The specific surface area of the sample obtained in Group C was 1.6494 ± 0.0134 m^2^/g, data similar to those obtained in Group A. 

#### 3.1.2. Bio-Oss

The xenograft was taken in monoblock format and crushed in the three ways described above. The following values were obtained for the specific surface area: Group A 18.2365 m^2^/g, Group B 25.2349 m^2^/g, and Group C 17.4529 m^2^/g.

### 3.2. Granulometry

The particle-size values of Group A were characterised by a large disparity between the particles obtained in terms of width and length. Table 1 shows the heterogeneity of the analysed particles. There was a large difference between the maximum and minimum values of particle lengths, and Table 2 shows how, with the width also, no heterogeneity between the particles was found, with a smaller discrepancy between the minimum and maximum.

Figure 3 and Figure 4 indicate that on the x-axis (x) are the sizes measured in micrometres (µm), and on the y-axis (y) are the volume or number of particles analyzed with these sizes. Most of the analysed particles were, on average, 866 µm long and 276 µm wide. 

Figure 5 shows four images of Group A taken by SEM, showing large versus smaller particles, demonstrating the large discrepancy in particle size obtained with the manual crushing method.

The average particle size obtained for Group B was 751.9 µm (Figure 6) and that of Group C was 828.1 µm (Figure 7).

Table 3 shows the distribution of the two groups according to the mean particle-size values obtained in EG.

The distinction between one size or the other is measured by the value of the volumetric diameter (DV), which is the size in micrometres of the sieve through which a % of the particles pass. D10 represents the sieve through which 10% of the smallest particles pass, D50 the sieve through which half of the sample is separated, and D90 reports the largest particle size. The uniformity will be given by the ratio DV (90/D10); the sample will be more uniform the smaller this ratio is. The values obtained for Group B were lower than those for Group C. This showed that Group B had a higher homogeneity than Group C because the D90/D10 ratio was lower for Group B, and the D90/D10 ratio was lower for Group B.

Table 4 shows the distribution of the three groups according to the mean particle size values obtained in CG. In this case, Group B also obtained lower values compared to Groups A and C. The D90/10 values were the lowest for Group B.

## 4. Discussion

The fine-grained Bio-Oss^®^ has a larger specific surface area, which enlarges the area of cell adhesion and promotes cell differentiation more efficiently than coarse-grained granules [59]. A higher quantity and quality of bone was formed after the use of small particles of bovine biomaterial compared to medium and large particles, which was justified by greater contact with the biomaterial surface and the surrounding tissues, thus producing greater bone neoformation [60]. The electric grinder that formed Group B presented the highest specific surface-area value (2.4025 ± 0.0218 m^2^/g), so the particles crushed by this method will offer a higher reactivity with the medium, unlike Groups A and C where the particles were more heterogeneous and larger in size with lower specific surface area values, 1.7451 ± 0.0183 m^2^/g and 1.6494 ± 0.0134 m^2^/g, respectively. 

The results obtained from CG with the three grinding methods described demonstrated that the highest value was obtained by Group B with values of 25.2349 m^2^/g as was the case with ATDG grinding, followed by Group A with 18.2365 m^2^/g and C with 17.4529 m^2^/g. This shows once again how the smart dentin grinder (Group B) presents higher specific surface-area values and, therefore, allows the result of the shredded biomaterial to be homogeneous with each other and, therefore, have greater contact with the biological surface on which it is deposited.

The size of the autologous bone particle was considered by several authors [61,62,63,64,65] as one of the main factors determining the ability of the graft to produce bone augmentation. Zaner and Yukna [63] assumed that small, densely packed particles could impede the migration and growth of cells, blood vessels, and bone cells.

Berberi et al. [66] performed an X-ray diffraction assay of the sample to be studied to determine the particle sizes of various groups to be analyzed, including Bio-Oss^®^. The DV (10) value for this xenograft was 1.32 µm and in DV (50–90) 0.26–8.92 µm, while the diameter described by the producers of the same was 250–1000 µm. 

In EG, Group B had a mean of 751.9 µm and in Group C 828.1 µm. The DV ratio (90/10) obtained lower values for Group B of 2.407 µm compared to Group C with 4.030 µm, which showed that the particles obtained with the electric grinder were much more homogeneous than those obtained with the manual grinder. Lower values for Group B were also obtained for GC, with a mean of 783.0 µm and a DV (90/10) value of 2.51 µm. This allows us to affirm once again that the electric grinder allows, in both cases, to generate more uniform particles.

When comparing the particle-size values obtained in the present study with the values available for Bio-Oss^®^, we observed that Group A had a mean value of 905.1 µm and 332.5 µm, Group B 751.9 µm, and Group C 828.1 µm. Bio-Oss^®^ is marketed in fine-grained (250–1000 µm) and coarse-grained (1000–2000 µm) formats. So, both Groups B and C fall within the values of the fine-grain group of Bio-Oss^®^. Group A reached a maximum particle length value of 2183 µm, so it could be classified into the coarse-grained group.

The particle-size results show that method B has the smallest particle size and the highest specific surface area. This is normal in powder materials, as seen in calcium phosphate particles [67,68]. The higher specific surface area means that the material obtained by methodology B will have a higher biological activity than A and C. As in the case of calcium phosphate cements regeneration materials with small size and high specific surface area, the conversion to bone is much faster than in the case of large particles. This fact will make it possible to control the regeneration rate [69].

It can also be seen that the particles obtained by method B have a lower dispersion than in the case of particles obtained by methods A and C. This fact is due to the fact that method B is automatic and, therefore, does not present the variations produced by the user. 

Kon et al. [70] concluded that the use of autologous fine-grained bone (150–400 µm) offered little possibility of bone augmentation and volume maintenance due to its rapid resorption, so it had to be combined with a slowly resorbing bone substitute despite its favourable osteogenic capacity. In contrast, coarse-grained bone (1000–2000 µm) showed an increase in volume with morphological stability over time but with limited osteogenesis, so it was considered as an option to combine it with fine-grained bone. For this reason, this same group of researchers in 2014 [70] combined coarse grain with fine grain, finally concluding that the combination of both offered no advantages in terms of maintaining bone volume. Similar results to those obtained by Klüppel et al. [62], who demonstrated that, sixty days after implantation of the fine-grained autologous graft (250–500 µm), there was almost complete resorption of the biomaterial. The increase in particle size (850–1000 µm) was beneficial in terms of resorption. Again, it was the fine-grained graft that was shown to promote greater bone neoformation compared to the coarse-grained graft. Higher mRNA levels of osteogenic markers were also observed in the fine-grained group when comparing Bio-Oss^®^ with the other existing biomaterials [61].

Scabbia and Trombelli [71] stated that Bio-Oss^®^ particles were larger (250–1000 µm) than those of synthetic hydroxyapatite of equine origin commercially called Biosites^®^ (Milan, Italy) with a smaller diameter (160–200 µm). Therefore, a size larger than 200 µm was considered to benefit the formation of a more solid physical scaffold, acting as a filler of the structural space and limiting the postoperative collapse of the supracrestal soft tissue of the defect.

However, there are limitations in this study, such as it may be that it has not been possible to analyze the granulometry of the three groups with the same apparatus due to the excessive size of the particle in Group A. In addition, subsequent in vivo studies will be needed to verify that Group B, which has obtained better properties, really offers better clinical results after applying this ATDG in the oral environment.

## 5. Conclusions

The sample obtained by the electric grinder had the highest value of specific surface area (2.4025 ± 0.0218 m^2^/g), while the particle size as average diameter (751.9 µm) was the lowest and most homogeneous of the three groups. Therefore, the electric grinder allowed for obtaining ATDG with more regenerative properties due to its specific surface area value and particle size in accordance with the xenograft with the greatest bibliographical support (Bio-Oss^®^). The higher specific surface increases the reaction with the physiological media producing faster biological mechanisms.

## Figures and Tables

**Figure 1 materials-17-00773-f001:**
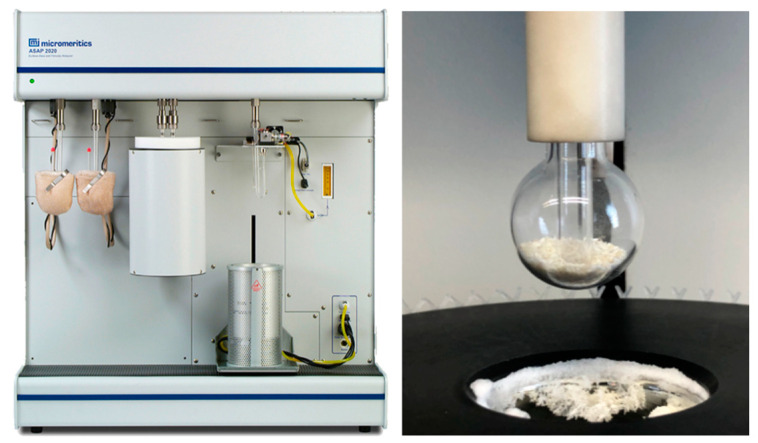
ASAP 2020 equipment.

**Figure 2 materials-17-00773-f002:**
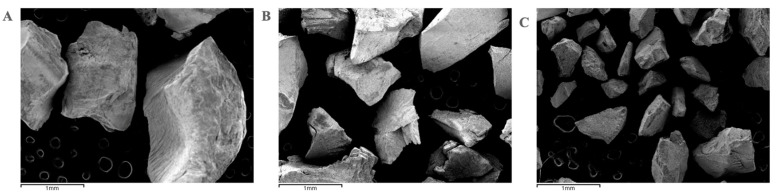
Representative SEM photographs of tooth ground with gouge (**A**), electric grinder (**B**), and hand grinder (**C**).

**Figure 3 materials-17-00773-f003:**
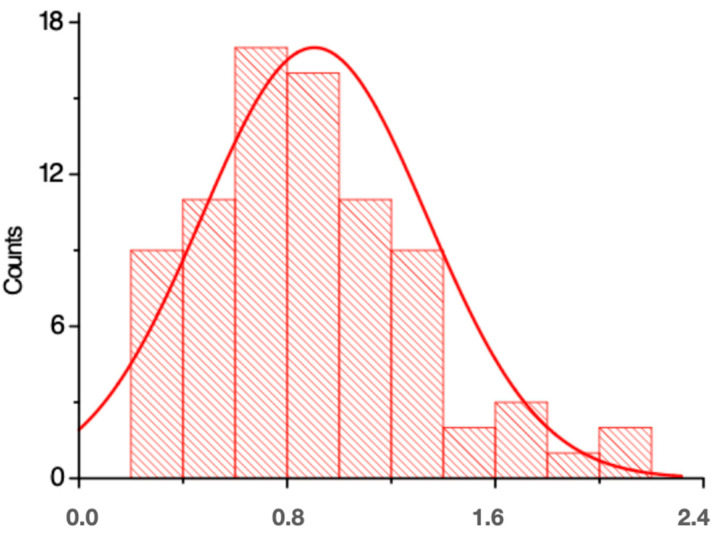
Particle length (mm) plot for Group A.

**Figure 4 materials-17-00773-f004:**
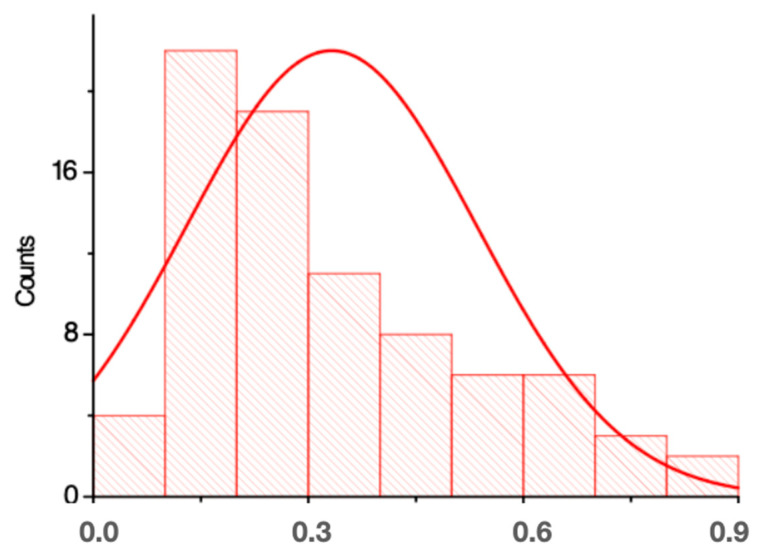
Plot of the particle width (mm) of Group A.

**Figure 5 materials-17-00773-f005:**
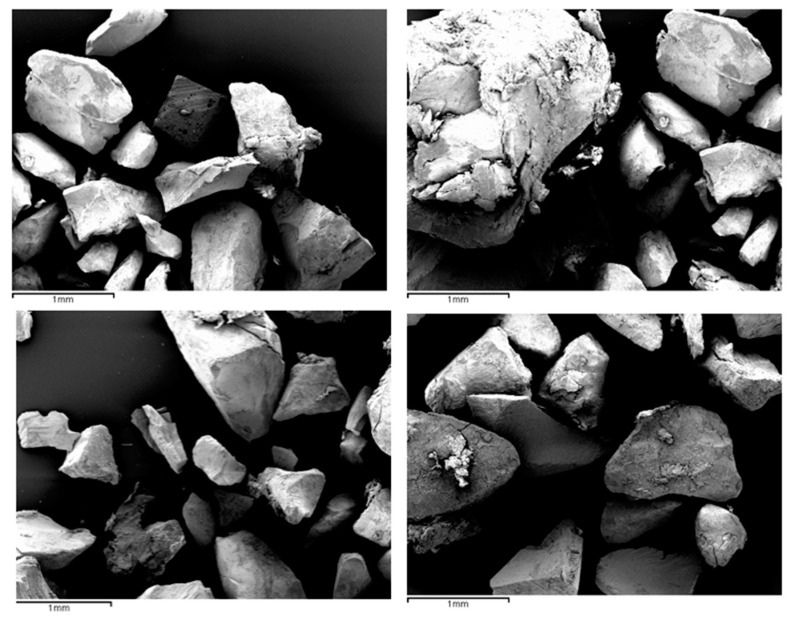
Images obtained with the SEM of Group A.

**Figure 6 materials-17-00773-f006:**
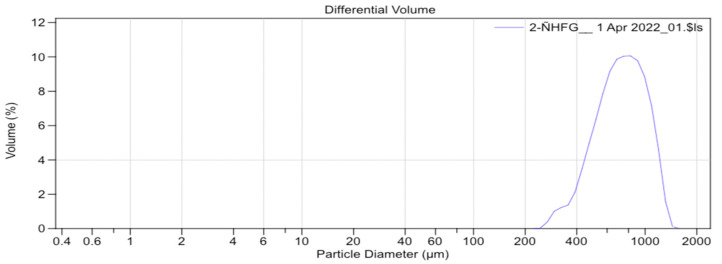
Particle-size distribution curve of the sample crushed with the electric grinder (Group B).

**Figure 7 materials-17-00773-f007:**
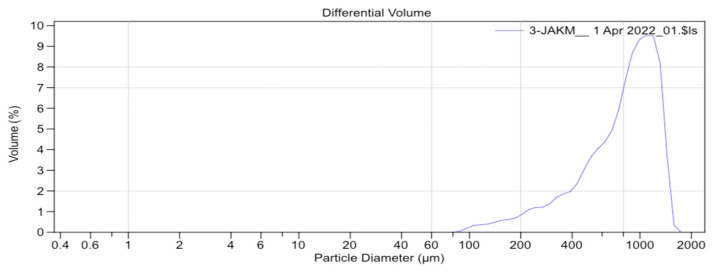
Particle-size distribution curve of the sample crushed with the manual grinder (Group C).

**Table 1 materials-17-00773-t001:** Particle length (µm) of Group A.

Total N	Mean	Standard Deviation	Minimum	Median	Maximum
81	905.11 µm	0.43342	21 µm	866 µm	2183 µm

**Table 2 materials-17-00773-t002:** Width (µm) of Group A.

Total N	Mean	Standard Deviation	Minimum	Median	Maximum
81	332.53 µm	0.20262	74 µm	276 µm	873 µm

**Table 3 materials-17-00773-t003:** Results of granulometry (µm) of the sample of EG.

Average Equivalent Diameter (µm)	Group B	Group C
Media	751.9 µm	828.1 µm
DV (10)	452.8 µm	319.6 µm
DV (50)	731.9 µm	856.5 µm
DV (90)	1090.0 µm	1288.0 µm
DV (90/10)	2.407 µm	4.030 µm

**Table 4 materials-17-00773-t004:** Results of granulometry (µm) of the sample of CG.

Average Equivalent Diameter (µm)	Group A	Group B	Group C
Media	987.2	783.0 µm	888.1 µm
DV (10)	232.3	502.8 µm	300.6 µm
DV (50)	1009.3	740.1 µm	896.4 µm
DV (90)	1559.2	1030.0 µm	1788.0 µm
DV (90/10)	6.712	2.051 µm	5.948 µm

## Data Availability

The data that support the findings of this study are available from the corresponding author upon reasonable request.

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
