# Peer review of "Autologous Tooth Granulometry and Specific Surface Area with Three Grinding Methods: An In Vitro Study"

_materials, 2024, doi:10.3390/ma17040773_

Round 1

Reviewer 1 Report

Comments and Suggestions for Authors

Dear author, the article is very premeeting, and the topic is very interesting in oral and maxillofacial surgery.

I am honored to share with you just few suggestions to try to improve the paper.

The present study is well designed and conducted, and the manuscript is quite clear.

The English is good and spelled correctly but to be published it needs of some adjustment.

There are some comments below.

Abstract: The abstract correctly summarizes the study design and purpose as the title as well.

Keywords: The keywords are correct and perfectly fitting the study design, I would add “dental extraction” if it is possible.

Introduction:

-  Line 31-43: You should improve your references adding also concepts and sentence to explain why this resorption is more in the fist part of the time after the extraction. The process of Bone loss is significant in the first six months post-extraction yes, but especially in the fist weeks of healing due to the granulation tissue and the contractile action of Myofibroblast how it was demonstrated in the in vitro studies such as the following one that you could mention.

Covani U, Giammarinaro E, Panetta D, et all. Alveolar Bone Remodeling with or without Collagen Filling of the Extraction Socket: A High-Resolution X-ray Tomography Animal Study. J Clin Med. 2022 Apr 29;11(9):2493. doi: 10.3390/jcm11092493.

The process of bone loss continues slowly throughout life because of the lack of dental vascularization and also the function of periodontal ligament is missing.

Moreover, it depends also by the type of surgical extraction and the disinfection of the alveolar site. 

About socket preservation in the introduction you could mention about this new autologous graft materials in the following article:

Minetti E, Celko M, Contessi M, Carini F, Gambardella U, Giacometti E, Santillana J, Beca Campoy T, Schmitz JH, Libertucci M, Ho H, Haan S, Mastrangelo F. Implants Survival Rate in Regenerated Sites with Innovative Graft Biomaterials: 1 Year Follow-Up. Materials (Basel). 2021 Sep 14;14(18):5292. doi: 10.3390/ma14185292.

-        Discussion

Moreover, you should report also other clinical articles contrasting what you reported.

Regarding the fact that it is very important to be atraumatic and minimally invasive as possible also in all extraction in dentistry to reduce bone resorption and aesthetic issue and to perform the socket preservation you should cite also the technique of customized abutment and without biomaterials, but simply managing the soft tissue with the Provisional prothesis not loaded. This Group of this article was one of the first for this technique in literature.

Menchini-Fabris GB, Crespi R, Toti P, Crespi G, Rubino L, Covani U. A 3-year retrospective study of fresh socket implants: CAD/CAM customized healing abutment vs cover screws. Int J Comput Dent. 2020;23(2):109-117. PMID: 32555764.

Anyway, the rest of article and conclusion are well done.

Please explain more about the limitation of this study in the discussion, also about the in vitro nature of the study because admitting the limitation is a good aspect of a single study bringing researcher to improve for future in vitro or clinical study.

I hope these suggestions may help you to publish the article.

Author Response

REVIEWER 1

Dear Reviewer,

Thanks for taking the time to review our manuscript and suggest to us to improve our work by providing a lot more detail. We have done so, and we are now submitting a manuscript that not only addresses the points you specifically raised but also many others that we have considered in order to deliver what we think is a much improved version of our work. This version includes more paragraphs, English grammar revisions in all main sections, Thanks a lot. We are looking forward to your comments.

Sincerely,

Javier Gil Mur

Keywords: The keywords are correct and perfectly fitting the study design, I would add “dental extraction” if it is possible. the word ''dental extraction'' has been added to the keywords

Introduction:

Line 31-43: You should improve your references adding also concepts and sentence to explain why this resorption is more in the fist part of the time after the extraction. The process of Bone loss is significant in the first six months post-extraction yes, but especially in the fist weeks of healing due to the granulation tissue and the contractile action of Myofibroblast how it was demonstrated in the in vitro studies such as the following one that you could mention.

Covani U, Giammarinaro E, Panetta D, et all. Alveolar Bone Remodeling with or without Collagen Filling of the Extraction Socket: A High-Resolution X-ray Tomography Animal Study. J Clin Med. 2022 Apr 29;11(9):2493. doi: 10.3390/jcm11092493.

The process of bone loss continues slowly throughout life because of the lack of dental vascularization and also the function of periodontal ligament is missing. Moreover, it depends also by the type of surgical extraction and the disinfection of the alveolar site. 

We greatly appreciate the proposal of this interesting article, after its analysis we did not find the data to support the concepts that you suggest us to include in the article, we thought that we could use the classic articles from the University of Göteborg to address that issue (G Cardaropoli, M Araújo, J Lindhe Dynamics of bone tissue formation in tooth extraction sites. An experimental study in dogs. J Clin Periodontol. 2003 Sep;30(9):809-18. doi: 10.1034/j.1600-051x.2003.00366.x. and Mauricio G Araújo, Jan Lindhe Dimensional ridge alterations following tooth extraction. An experimental study in the dog J Clin Periodontol. 2005 Feb;32(2):212-8. doi:10.1111/j.1600-051X.2005.00642.x.) that le would seem to use these items?

About socket preservation in the introduction you could mention about this new autologous graft materials in the following article:

Minetti E, Celko M, Contessi M, Carini F, Gambardella U, Giacometti E, Santillana J, Beca Campoy T, Schmitz JH, Libertucci M, Ho H, Haan S, Mastrangelo F. Implants Survival Rate in Regenerated Sites with Innovative Graft Biomaterials: 1 Year Follow-Up. Materials (Basel). 2021 Sep 14;14(18):5292. doi: 10.3390/ma14185292.

The reference of the article has been added to the paragraph where the use of ATDG as regenerative material is discussed.

Discussion

  • Moreover, you should report also other clinical articles contrasting what you reported. Currently there are no scientific articles that talk about the same topic as this study, which is why it was decided to carry out a study on it.
  • Regarding the fact that it is very important to be atraumatic and minimally invasive as possible also in all extraction in dentistry to reduce bone resorption and aesthetic issue.

     If you agree, we can refer to the following article:

  • And to perform the socket preservation you should cite also the technique of customized abutment and without biomaterials, but simply managing the soft tissue with the Provisional prothesis not loaded. This Group of this article was one of the first for this technique in literature. Menchini-Fabris GB, Crespi R, Toti P, Crespi G, Rubino L, Covani U. A 3-year retrospective study of fresh socket implants: CAD/CAM customized healing abutment vs cover screws. Int J Comput Dent. 2020;23(2):109-117. PMID: 32555764.

The information provided in the article about the provisionalization of the implant has been added in order to maintain the emergency profile

Please explain more about the limitation of this study in the discussion, also about the in vitro nature of the study because admitting the limitation is a good aspect of a single study bringing researcher to improve for future in vitro or clinical study.

A final paragraph was added to the discussion where the limitations of the present study are considered.

Reviewer 2 Report

Comments and Suggestions for Authors

The manuscript titled “Autologous tooth, granulometry and specific surface area with three grinding methods: an in-vitro study” aimed to evaluate the most relevant parameters to achieve the best properties of ground autologous tooth-derived graft using three different methods. While the reviewer appreciates authors’ efforts for having a clear objective and straightforward outcomes, further clarification and presentation are paramount to strengthen the impact and readability of the study for possible publication in materials.

Comments:

1. As the authors mentioned during the Introduction, they used the electrical grinder with specific settings – “The grinding time was three cycles of twenty seconds each at 240 Volt (V), with a frequency of 50 Herz (Hz) at 16000 revolution per minute (rpm).” It would definitely enrich the study to explore or discuss how variations in these parameters (e.g., longer grinding time or increased speed) might affect the properties of the ground autologous tooth-derived graft. Please provide additional evidence or a more detailed discussion on this aspect. (major).

2. Please re-arrange Figure 2 to ensure that panel A, B, and C are of the same size, magnification and include a uniform scale bar to facilitate a direct comparison for reviewers and readers. (Minor)

3. It is confusing to have both Figure 2 and figure 5 showing the SEM images. What’s the difference from these Fig.2 and Fig.5? And what’s the point showing four SEM images for the group A (manual grinding) in Fig. 5. Please consider combine Fig. 2 and Fig. 5 fore more streamlined presentation. (major)

4. Figure 3 and Figure 4 appear to be the direct snapshots from Word or Excel. Please provide higher quality images. And the x-axis for Figure 4 should be labeled as “Particle Width” instead of “particle length”. (minor)

5. Figure 6 and Figure 7 are both particle size distribution curves from the samples crushed with the electric grinder or manual grinder. It is confusing that why the particle size distribution curve is not shown for the sample processed by the gouge forceps (group A)? Please provide this data and consider combining these three datasets together into one figure to allow for a more effective comparison. (Major)

Comments on the Quality of English Language

The quality of English language is satisfactory.

Author Response

REVIEWER 2

Dear Reviewer,

Thanks for taking the time to review our manuscript and suggest to us to improve our work by providing a lot more detail. We have done so, and we are now submitting a manuscript that not only addresses the points you specifically raised but also many others that we have considered in order to deliver what we think is a much improved version of our work. This version includes more paragraphs, English grammar revisions in all main sections, Thanks a lot. We are looking forward to your comments.

Sincerely,

Javier Gil Mur

It would definitely enrich the study to explore or discuss how variations in these parameters (e.g., longer grinding time or increased speed) might affect the properties of the ground autologous tooth-derived graft. Please provide additional evidence or a more detailed discussion on this aspect. (major).

Regarding the voltage, rpm and hertz used in the study, it was indicated by the manufacturer of the electric grinder. If the potential were lower, the power would not fracture the particles and the sizes would be larger than those obtained with correct crushing. If the potential increases, it would reduce its size further, but excess power could denature the material due to the great power used. Furthermore, the electric grinder is already configured to have these properties so they could not be modified even if we wanted to.

  1. Please re-arrange Figure 2 to ensure that panel A, B, and C are of the same size, magnification and include a uniform scale bar to facilitate a direct comparison for reviewers and readers. (Minor)

The images in Figure 2 have been modified, showing an example of each of the three groups. All images were captured at 35 magnification. This has also been specified in the text

  1. It is confusing to have both Figure 2 and figure 5 showing the SEM images. Whats the difference from these Fig.2 and Fig.5? And whats the point showing four SEM images for the group A (manual grinding) in Fig. 5. Please consider combine Fig. 2 and Fig. 5 fore more streamlined presentation. (major)

In the figure 2 its represented a sample of the 3 grinding methods while in fig. 5 we wanted to show the differences between the particles that the manual method results in by showing it with 4 images of this method

  1. Figure 3 and Figure 4 appear to be the direct snapshots from Word or Excel. Please provide higher quality images. And the x-axis for Figure 4 should be labeled as Particle Widthinstead ofparticle length. (minor)

Images 3 and 4 have been introduced with higher quality in the paper

  1. Figure 6 and Figure 7 are both particle size distribution curves from the samples crushed with the electric grinder or manual grinder. It is confusing that why the particle size distribution curve is not shown for the sample processed by the gouge forceps (group A)? Please provide this data and consider combining these three datasets together into one figure to allow for a more effective comparison. (Major)

Thank you very much, very good recognition, it was impossible for us to measure the three groups with the same device, groups B and C were made using laser particle size analyser Beckman Coulter®, while group A was made using the scanning electron microscope (SEM), the reason was that the size of some of the group A particles was too large to be measured by the other machine. We tried to explain in the article with the text ''The particle size analysis of the samples from the crushed group with the manual and electric grinder was carried out using a Beckman Coulter® (California, USA) model LS13 320 laser particle size analyser using the "Tornado" configuration and dry analysis, with a particle diameter measurement range between 0.04–2000 µm. The Malvern Panalytical® Mastersizer 3000 liquid analysis equipment was not used due to the impossibility of adjusting the obscuration of the liquid medium between 5% and 10% recommended by the equipment manufacturer and due to the problems of obstruction of the equipment's aspiration ducts as it was not possible to measure diameters greater than 1 mm. For these reasons, the samples were analysed with the dry analysis equipment.This was carried out using a laser diffraction technique to measure particle size by measuring the intensity of scattered light as a laser beam passed through the particulate sample. This data was analysed to calculate the size of the samples according to the scattering pattern.Since group A presented values that the high-sensitivity particle size equipment did not admit because they were particles larger than 2000 μm, the sample had to be analyzed randomly with a JEOL JSM 5410® scanning electron microscope (SEM) (Musashino, Akishima, Tokyo) performing a total of three thousand measurements of eighty-one particles with the Image J image analysis system (Tokyo, Japan) integrated with the SEM, between which the sample was shaken and deposited back on the plate to perform a measurement of each particle, obtaining an average of all the measurements performed and thus a value for its particle size’’  but if you consider necessary to write it in another way to make it clearer we can try it. In the same way we can to fuse the graph of group b and c in the same image if you deem it appropriate.